# Organic resolution function and effects of platinum nanoparticles on bacteria and organic matter

**Hiroo Itohiya**◉, **Yuji Matsushima**◉, **Satoshi Shirakawa, Sohtaro Kajiyama, Akihiro Yashima, Takatoshi Nagano, Kazuhiro Gomi**◉ *

Department of Periodontology, Tsurumi University, School of Dental Medicine, Tsurumi, Tsurumi ku, Yokohama, Japan

◉ These authors contributed equally to this work.
* gomi-k@tsurumi-u.ac.jp

**Data Availability Statement:** All relevant data are within the paper and its Supporting Information files.

**Funding:** The authors received no specific funding for this work.

## Abstract

Rapid progress has been made in terms of metal nanoparticles studied in numerous fields. Metal nanoparticles have also been used in medical research, and antibacterial properties and anticancer effects have been reported. However, the underlying mechanism responsible for these effects has not been fully elucidated. Therefore, the present study focused on platinum nanoparticles (PtNPs) and examined their antibacterial properties and functional potential for decomposing organic matter, considering potential applications in the dental field. PtNPs were allowed to react with dental-related bacteria (*Streptococcus mutans*; *Enterococcus faecalis*, caries; *Porphyromonas gingivalis*, and endodontic and periodontal lesions). Antibacterial properties were evaluated by measuring colony formation. In addition, PtNPs were allowed to react with albumin and lipopolysaccharides (LPSs), and the functional potential to decompose organic matter was evaluated. All evaluations were performed *in vitro*. Colony formation in all bacterial species was completely suppressed by PtNPs at concentrations of >5 ppm. The addition of PtNPs at concentrations of >10 ppm significantly increased fragmentation and decomposition. The addition of PtNPs at concentrations of >125 pico/mL to 1 EU/mL LPS resulted in significant amounts of decomposition and elimination. The results revealed that PtNPs had antibacterial effects against dental-related bacteria and proteolytic potential to decompose proteins and LPS, an inflammatory factor associated with periodontal disease. Therefore, the use and application of PtNPs in periodontal and endodontic treatment is considered promising.

## Introduction

Metal nanoparticles, which are obtained by converting metals into fine particles (< 100 nm in diameter), have a large surface area and exhibit properties that differ from those of bulk metals because of the quantum size effect [1]. Metal nanoparticles have been studied in numerous fields, such as chemistry, biology, materials science, and medicine, and rapid advances have

**Competing interests:** The authors have declared that no competing interests exist.

been reported in recent years [2–6]. The antibacterial characteristics of metal nanoparticles (silver, zinc oxide, copper oxide, etc.) have previously been reported [7–12]. In particular, silver nanoparticles (AgNPs) have attracted much interest in many fields due to their excellent broad-spectrum antibacterial activity [9,13]. In contrast, gold nanoparticles (AuNPs) are commonly evaluated for use in biosensing or drug delivery applications, given that these particles are inert and highly stable [14]. Antibacterial AuNPs are produced by coating AuNPs with organic molecules that have antibacterial properties [15].

Previous studies have reported the antibacterial effects of platinum nanoparticles (PtNPs). For example, PtNPs slow cell division in *Escherichia coli* [16]. Furthermore, PtNPs, which are clusters of Pt atoms in the range of 1–100 nm, are attracting attention due to the extremely high catalytic activity. Recently, the ability of these nanoparticles to suppress inflammation has also been evaluated [17,18]. Chemical interactions are thought to be induced by contact between bacteria and PtNPs, leading to the decomposition of bacteria. This is believed to trigger a chain reaction in which free radicals produced from the antioxidant response of PtNPs damage bacteria [19]. While one study has found that PtNPs increase the level of intracellular reactive oxygen species (ROS) [20], another study has reported that PtNPs eliminate ROS [21].

In the present study, we focused on the application of metal nanoparticles in the dental field because platinum is a precious metal that is unlikely to cause allergies, does not demonstrate genotoxic potential, and has high potential for clinical application [15].

Although metal nanoparticles possess various useful properties, nanoparticles clump into a large aggregate when left in their original state, altering their properties. To maintain the stable dispersed state of nanoparticles, protective agents are generally added to adsorb and configure on the surface of metal nanoparticles and to prevent nanoparticle aggregation [22]. However, the use of protective agents reportedly hinders expression of the full range of properties that metal nanoparticles originally possess [23].

In this study, we used PtNPs, which were produced directly from platinum by irradiating an infrared pulsed laser according to the Liquid phase laser ablation method [24]. The aim of this study was to investigate the functional potential of nanoparticles (PtNPs) to decompose organic matter and antibacterial activity against dental-related bacteria, which are important indicators of the potential application of PtNPs in the field of dentistry.

## Materials and methods

In this experiment, 100 ppm PtNPs (particle size: 2–19 nm) solution, which was used as a stock solution, were obtained from the manufacturer (Ceramics Craft Co. Ltd. Shizuoka, Japan). This PtNPs were made directly from platinum by irradiation with infrared pulsed laser in liquid. The storage of PtNPs was kept in the dark at room temperature avoiding direct sunlight as the stock solution. Moreover, when using for experiment, the stock solution was diluted in sterile purified water and used each time. The PtNPs were examined by transmission electron microscope (JEM-1200EX II, JEOL, Tokyo, Japan) Fig 1 shows a transmission electron micrograph of PtNPs used in this experiment.

### Measurement of antibacterial activity

**Bacterial culture and regulation.** Experiments were performed with bacteria that are important in dentistry, namely the *Streptococcus mutans* ATCC25175 strain, *Enterococcus faecalis* ATCC19433 strain, and *Porphyromonas gingivalis* ATCC 33277 strain. *Streptococcus mutans* is a gram-positive bacterium that causes caries; *E. faecalis* is a gram-positive bacterium that causes refractory apical periodontitis; and *P. gingivalis* is a gram-negative obligate anaerobic and a periodontal pathogenic bacterium. All bacteria were purchased from American Type

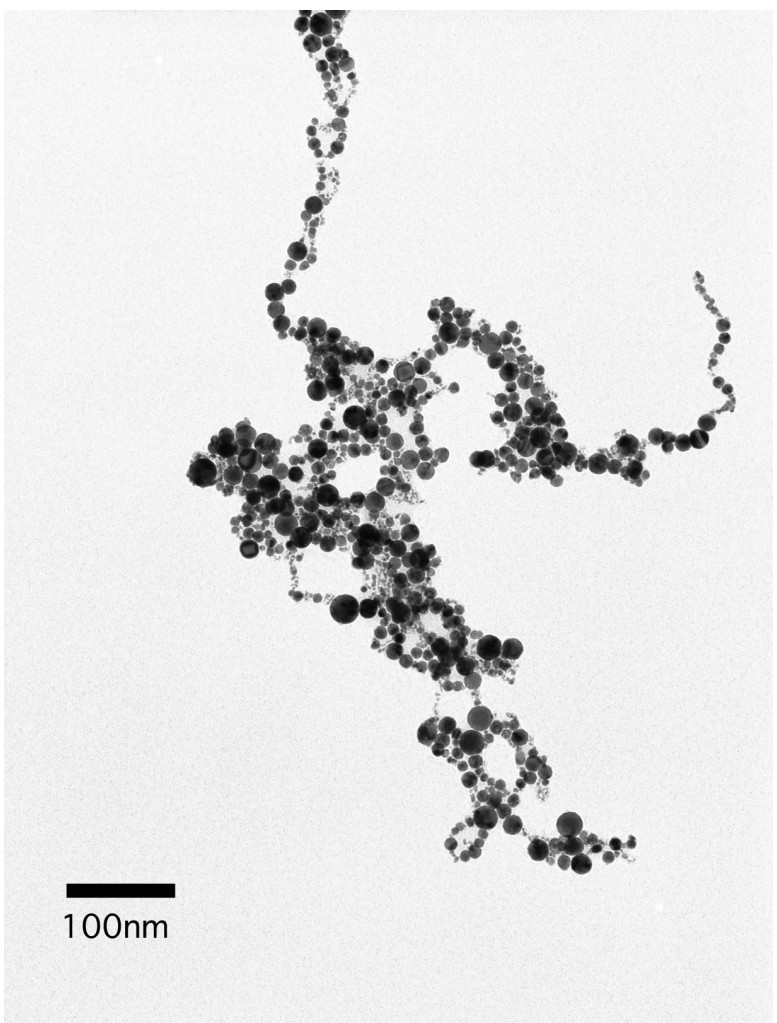

**Fig 1. Transmission electron micrograph of PtNPs.** 10 ppm of PtNPs, which diluted by pure water were observed at 80kv 45μA. (scale bar = 100 nm) Nearly the same size of PtNPs as the manufacturer indicated (2-19nm) were observed.

Culture Collection (ATCC), VA, USA. *Streptococcus mutans* was cultured in Tryptic Soy Broth (TSB, FUJIFILM Wako Pure Chemical Co., Osaka, Japan) and agar medium (Becton Dickinson) with 4% glucose (FUJIFILM Wako Pure Chemical Co.) at 37°C under aerobic conditions at 100 rpm in shaking incubator (SI-782, Optima Inv. Tokyo, Japan) for 2 days, followed by preculture in TSB liquid medium with 4% glucose under aerobic shaking for 2 days. *Enterococcus faecalis* was cultured in TSB agar medium (at 37°C under aerobic conditions for 2 days), followed by preculture in TSB liquid medium under aerobic shaking at 100 rpm in shaking incubator for 2 days. *Porphyromonas gingivalis* was anaerobically cultured in an anaerobic culturing device (ANX-1, Hirasawa, Tokyo, Japan) under $N_2$: 80%, $CO_2$: 10%, and $H_2$: 10% conditions at 37°C for 1 week.

Precultured *S. mutans*, *E. faecalis*, and *P. gingivalis* were centrifuged for 5 minutes at 4°C and 7,000 ×g, harvested, and suspended in distilled water. Bacterial counts were adjusted to $10^5$–$10^6$ CFU/mL at an optical density of 620 nm with a spectrophotometer (UV-1200, SHIMAZU Co., Kyoto, Japan).

**Antibacterial testing.** PtNPs were added to 100 μl of $1 \times 10^4$ CFU/mL bacterial solutions to achieve final concentrations of 1, 5, 10, and 20 ppm. Samples that had been allowed to react for 10 minutes were inoculated onto bacterial culture medium [25]. The associated antibacterial effects were determined based on the presence or absence of colony formation on the medium. The group without PtNPs in the bacterial medium was defined as the positive control group, and the group to which lipopolysaccharides (LPSs) were not added was defined as the negative control group.

## Measurement of functional potential for organic resolution

**Electrophoresis and BCA protein assay.** To investigate the functional potential for organic resolution of PtNPsolutions, PtNPsolutions were added to albumin suspensions and were allowed to react (experimental groups), and the state of decomposition was observed. PtNPsolutions (100 ppm) were added to 2 mg/mL albumin aqueous solution to achieve a final concentration of 10 ppm. The mixture was stirred and allowed to react at room temperature for 1, 10, 30, and 60 minutes. In addition, suspension with only albumin was defined as the control group. Functional potential for organic resolution was evaluated through protein abundance measurement using polyacrylamide electrophoresis and the bicinchoninic acid (BCA) assay.

**Electrophoresis assay.** Samples collected from the experimental and control groups were electrophoresed using 15% polyacrylamide containing 1% sodium dodecyl sulfate (SDS) according to the Laemmli method [26]. In other words, each sample was dissolved in 1 mL of 0.01 mol/l Tris-HCL buffer solution (pH 6.8) containing 1% SDS and 25% glycerin to be used as samples for electrophoresis. After electrophoresis at 40 mA for 80 minutes under a low-voltage condition using the electrophoresis tank (Pagel AE-600, SPS-15S, ATTO Co., Japan), staining was performed with Coomassie brilliant blue (CBB, Bio-Rad, USA), followed by decoloration with stain/destain solution (Bio-Rad), and bands were detected.

**BCA assay.** The amount of albumin in each collected sample was measured using the BCA assay [27]. Overall, 200 μl of each sample was dispensed to a 96-well multiplate, allowed to react for 15 minutes according to the manual of the protein assay kit (Pierce Protein assay kit, Thermo SCIENTIFIC), and absorbance was measured (E-MAX Plus Microplate reader, Molecular Devices Co., Japan) at 562 nm. In this bacterial culture assay, the only standard was a sample with no bacterial growth at a concentration of 10 ppm.

**Measurement of ability to decompose lipopolysaccharides.** We studied whether LPS, a pathogenic factor secreted by some periodontal bacteria, could be decomposed by PtNPs. PtNPs were allowed to react with LPS, and the amount of LPS was measured. The Toxin Sensor Chromogenic LAL Endotoxin Assay Kit (Gen Script Co., USA) was used in the measurement of LPS. PtNPsolutions at 0, 10, or 50 ppm were added to 0.625, 1.25, 2.5, 5.0, and 10.0 EU/mL LPS. The mixture was allowed to react in a tube placed at room temperature for 10 minutes. The amount of LPS was measured by measuring the absorbance at 545 nm. Furthermore, to investigate the effective concentration of PtNPs against 1 EU/mL of LPS, a serially diluted PtNPsolution (0.015625–1 ppm) was added. The mixture components were then allowed to react for 10 minutes, at which point absorbance was measured, and LPS was detected.

## Statistical analysis

In the colorimetric determination of albumin and LPS decomposition ability, obtained values were subjected to one-way analysis of variance (one-way ANOVA) and Tukey's multiple

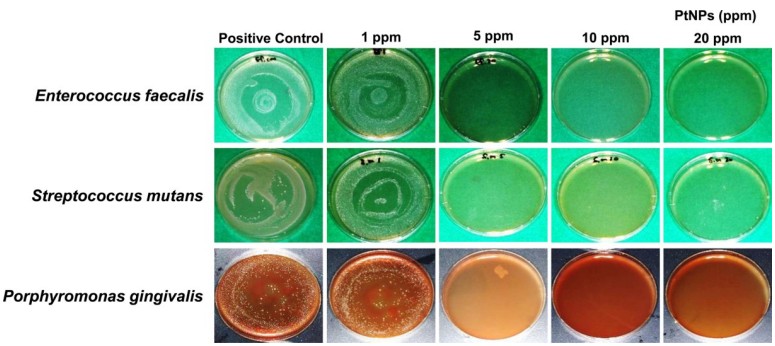

**Fig 2. Antibacterial activity of nanoplatinum.** The nanoplatinum aqueous solutions (concentrations of 1, 5, 10, and 20 ppm) were added to bacterial solutions of $1 \times 10^4$ *Streptococcus mutans*, *Enterococcus faecalis*, or *Porphyromonas gingivalis* and then allowed to react for 10 minutes. At concentrations up to 5 ppm, colony formation was completely suppressed in all bacterial strains.

comparison tests, with significance levels for both set at 0.05. All statistical analyses were performed using SPSS Statistics Version 11.5 (IBM Japan).

# Results

## Antibacterial activity of PtNPsolutions

To investigate the antibacterial activity of PtNPs, PtNPs were added to *S. mutans*, *E. faecalis*, and *P. gingivalis* bacterial solutions to achieve concentrations of 1, 5, 10, and 20 ppm. Samples that had been allowed to react for 10 minutes were inoculated onto each medium, and the presence or absence of colonies was confirmed to determine the antibacterial effect. Colony formation was completely suppressed in all bacterial strains at concentrations of >5 ppm. However, at a concentration of 1 ppm, colonies exhibited growth equivalent to that in the negative control group (which had not been treated with PtNPs). The same trend was observed for all bacterial strains investigated in this study (Fig 2).

## Functional potential for organic resolution of PtNPs

**Albumin decomposition.** To investigate the functional potential of PtNPs for organic resolution, albumin suspensions were allowed to react with PtNPs. The results were visualized using polyacrylamide electrophoresis. A comparison between the bands of control albumin and albumin that had been exposed to PtNPs in the electrophoretic image revealed a time-dependent decrease in the bandwidth of albumin in the experimental groups (Fig 3). In addition, smear layers believed to be albumin decomposition products were observed beneath the bands.

The measurement of protein abundance using the BCA protein assay revealed findings in line with those observed in the electrophoretic image. When compared with the absorbance in the control group, the absorbance of the experimental groups, which had been exposed to PtNPs, decreased significantly ($p < 0.05$) after 1 minute of activity and became almost constant after 30 minutes (Fig 4).

The electron micrograph of PtNPs mixed with albumin is shown in Fig 5A. There was no morphological change in PtNPs.

**LPS decomposition.** To study LPS decomposition, PtNPs (10 and 50 ppm) were allowed to react with LPS adjusted to 10.0, 5.0, 2.5, 1.25, or 0.625 EU/mL for 10 minutes. At 10 minutes, the amount of LPS was measured. The results showed that absorbance decreased

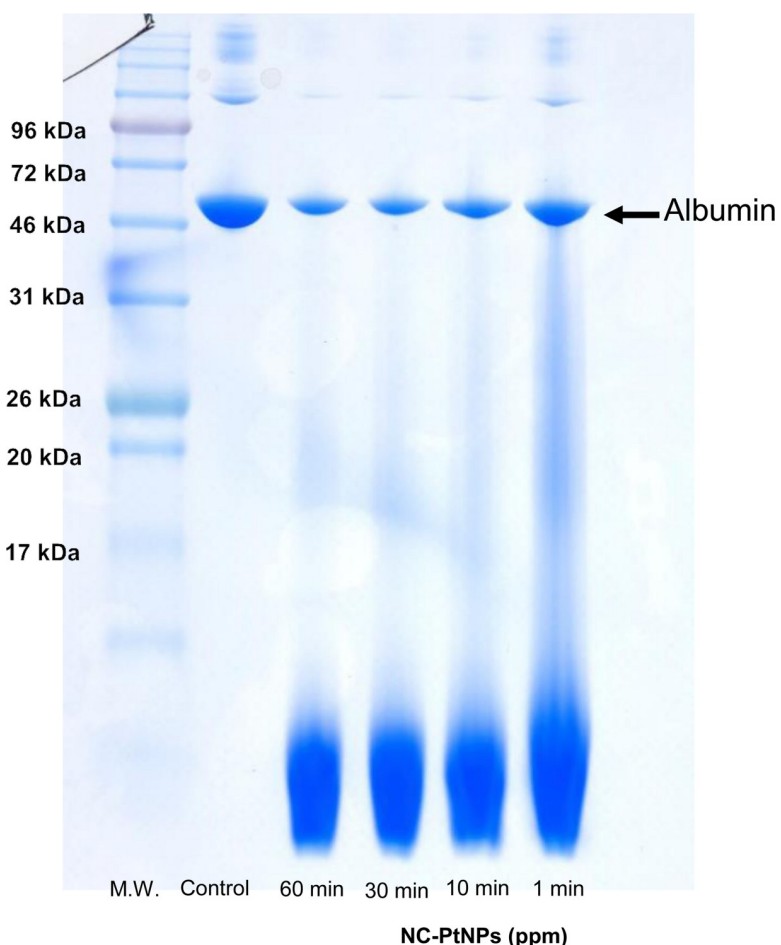

**Fig 3. Functional potential of nanoplatinum solutions to decompose albumin.** Overall, 10 ppm nanoplatinum solutions were added to 2 mg/mL albumin suspension, and the state of samples, which had been allowed to react for 1, 10, 30, and 60 minutes, was confirmed using electrophoresis. A time-dependent decrease in bandwidth was observed, and smear layers believed to be albumin decomposition products were observed beneath the bands. (M.W.: molecular weight).

significantly with the addition of PtNPs in all groups (Fig 6). Next, to study the minimum effective concentration of PtNPs, serially diluted solutions of 1 ppm PtNPs were added to 1 EU/mL LPS. The mixture was allowed to react for 10 minutes. The results showed that LPS was significantly degraded ($p < 0.05$) compared with the control at a concentration of 0.125 ppm PtNPs. At concentrations of >0.25 ppm, LPS values equivalent to those of the negative control group, which had not been exposed to LPS, were confirmed (Fig 7). The electron micrograph of PtNPs mixed with LPS is shown in Fig 5B. There was no morphological change in PtNPs.

## Discussion

Aqueous solutions of nano-sized metal particles reportedly exhibit more potent antibacterial activity as particle size enters the nanometer range [28]. In particular, AgNPs are known to have excellent antibacterial effects, but they are also known to possess strong cytotoxicity [29]. Recently, metal NPs have attracted much attention not only due to their antibacterial properties but also because of their application in other areas of medicine, such as cancer treatment

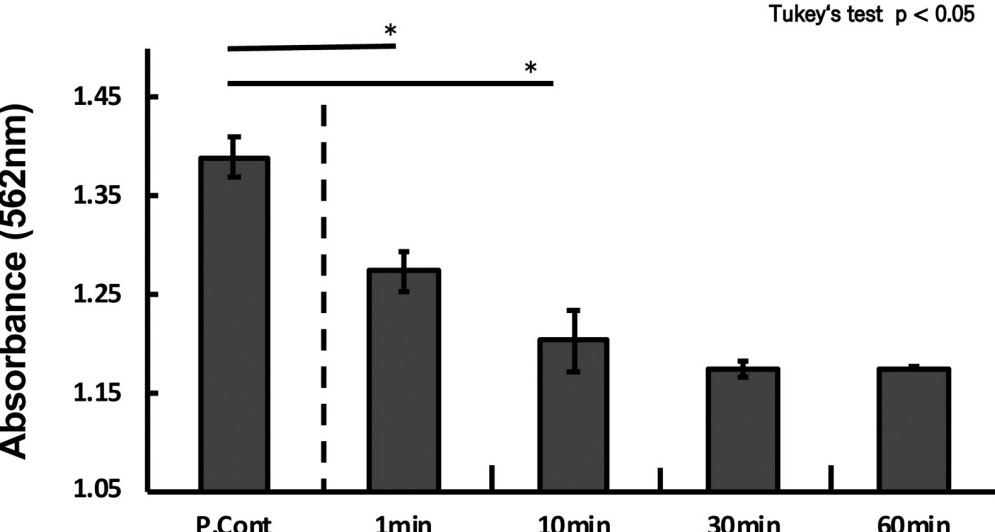

**Fig 4. Functional potential of nanoplatinum solutions to decompose albumin. Overall, 10 ppm nanoplatinum solutions were added to 2 mg/mL albumin solutions and allowed to react for 1, 10, 30, and 60 minutes.** Protein abundance was measured in the BCA protein assay. ($^*$p $<$ 0.05, Tukey's test).

[30]. This study focused on platinum, a stable metal that does not tend to induce allergic reactions, and investigated the effects of platinum nanoparticles. In this study, we used the PtNPs, which were made directly from platinum by irradiation with infrared pulsed laser in liquid according to the Liquid phase laser ablation method [24]. When PtNPs solutions were added to *S. mutans*, *E. faecalis*, and *P. gingivalis* adjusted to $1 \times 10^4$ cells/mL, clear antibacterial properties were observed at concentrations of $>$ 5 ppm. In a similar experiment that was previously reported, nanoplatinum particles (2–19 nm) were allowed to react for 15 minutes. These nanoparticles exhibited the same levels of antibacterial activity as that reported in our study, even

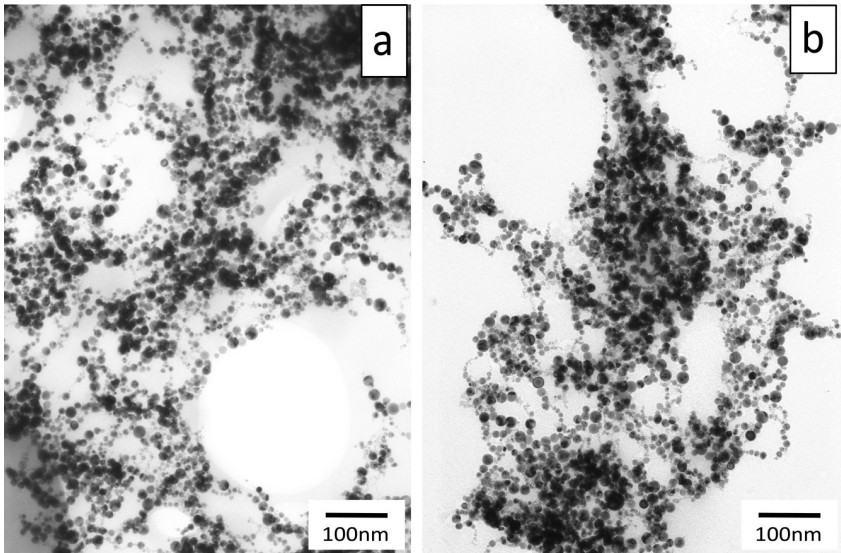

**Fig 5. The electron micrograph of PtNPs mixed with albumin and LPS. a): 10ppm PtNPs with 2 mg albumin, b) 10ppm PtNPs with LPS.** (scale bar = 100 nm,) PtNPs were distributed to adhere to albumin or LPS.

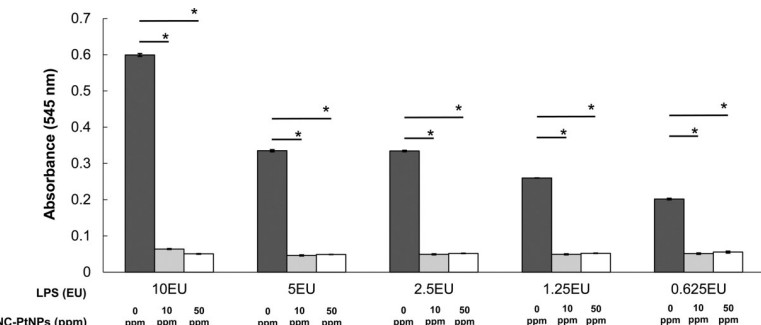

**Fig 6. Functional potential of nanoplatinum solutions to decompose LPS.** Overall, 10 and 50 ppm PtNPs were added to LPS at different concentrations and allowed to react for 10 minutes. LPS decreased in all groups. (* $p < 0.05$, Tukey's test).

though the bacterial strains were different [31,32]. The reaction time of the solution used and platinum particle size were almost the same, and the results obtained were similar to those reported in our study. Kanieczny et al. conducted a study to compare the antibacterial properties against gram-positive and gram-negative bacteria using PtNPs. The results showed a stronger antibacterial effect against gram-negative bacteria [33] because of the thin cell wall of gram-negative bacteria. In addition, Rosenberg et al. have also reported on antibacterial activity against *E. coli*, a gram-negative bacterium [16]. In this study, PtNPs demonstrated strong antibacterial activity not only against *P. gingivalis*, a gram-negative bacterium, but also against *S. mutans* and *E. faecalis*, which are gram-positive bacteria. While, it is inferred that the antibacterial effect of NPs on anaerobic bacteria of oral origin is lower than that on aerobic bacteria. It is considered that this is because the antibacterial property is controlled by the availability of oxygen and the particle size [34]. These interactions may have shown similar antibacterial activity to *P. gingivalis*, *S. mutans* and *E. faecalis*.

A possible mechanism for antibacterial effects includes damage to the cell wall or cell membrane, leading to the outflow of cytoplasm and invasion of NPs into the cell. NPs bind with proteins, stopping cell function and ultimately killing the bacterium [10]. Through transmission electron microscopic images, Chwalibog et al. have shown that PtNP-mediated damage to bacterial cell walls allows cytoplasm to flow out of the cell [35,36]. However, the mechanism by

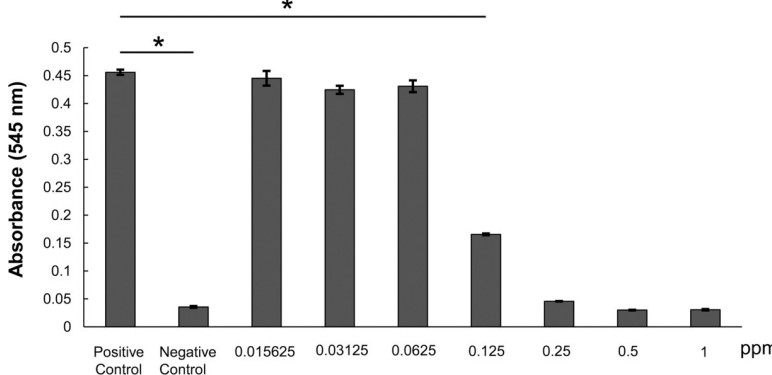

**Fig 7. Effective concentration of nanoplatinum solution against LPS.** PtNPs at different concentrations were added to 1 EU/mL LPS and allowed to react for 10 minutes. At concentrations of >0.25 ppm, LPS was decomposed to a level equivalent to that in the negative control group, where no LPS had been added. In the positive control groups, LPS without nanoplatinum solutions was added. (*$p < 0.05$, Tukey's test).

which metal NPs damage cell walls and invade the bacterium has not been fully elucidated. However, if cell walls and cytoplasm, which comprise proteins, are damaged, PtNPs may exhibit proteolytic effects. We therefore investigated whether PtNPs possessed functional potential to decompose organic matter.

The results showed a time-dependent decrease in the bandwidth of albumin in the group where PtNPs (10 ppm) had been exposed to albumin (2 mg/mL) compared with experiments in which albumin was electrophoresed alone. Although albumin may have been adsorbed to PtNPs, a smear layer believed to be the decomposition products of albumin appeared beneath the band after 1 minute of reaction time. The bandwidth of albumin decreased as the reaction time increased, and the smear layers underneath gradually disappeared. Albumin was therefore considered to have decomposed in a time-dependent manner. In addition, protein abundance was measured using the BCA protein assay with the same samples. The results obtained up to the 30-min timepoint showed a significant time-dependent decrease in the amount of proteins in the experimental groups compared with the control group; protein levels did not decrease after 30 minutes. The decrease in bandwidth in the electrophoretic image, as well as the time-dependent decrease in the amount of protein revealed by the results of the BCA protein assay, indicates the decomposition of albumin by PtNPs.

Next, we studied the effects of PtNPsolution on LPS, a constituent of gram-negative bacterial cell walls. PtNPs were added to 0.625–10.0 EU LPS to achieve final concentrations of 10 and 50 ppm. The mixture was allowed to react for 10 minutes. The results showed that LPS decreased significantly in all experimental groups compared with the control group. LPS, which is a component of the cell wall of gram negative bacteria, is negatively charged and has a high affinity to NPs and binds tightly. And it is thought that destruction of LPS arises from LPS and NPs from a physical interaction [37]. Next, to detect the minimum effective concentration of PtNPs required to decompose 1 EU/mL LPS, PtNPs at concentrations of 0.01561–1 ppm were allowed to react for 10 minutes. The amount of LPS remaining was then measured. PtNP concentrations of up to 0.25 ppm showed almost the same values as those of the negative control group, suggesting LPS decomposition. Conversely, at concentrations of ≤0.0625 ppm, no significant difference was noted in the positive control group. Albumin and LPS in contact with PtNPs are thought to be reduced in molecular weight due to cleavage of the protein backbone due to the oxidative degradation reaction or physical interaction caused by the generated ROS [38]. However, the detailed mechanism is still unknown.

Periodontal diseases, which are prevalent worldwide, stem from infection with periodontal pathogens, which are typically gram-negative bacteria [39]. In addition, even after the periodontal pathogens die, LPS, a bacterial constituent, remains on the root surface, acts as an endotoxin, and causes inflammation in periodontal tissues [40]. The PtNPs used in this study were of a particle size of 2–19 nm and were particles with exposed metal surfaces. These NPs can decompose organic matter (especially LPS) effectively, even when used at extremely low concentrations. Therefore, PtNPs may be an effective tool for periodontal treatment by acting as antibacterial agents against periodontal pathogenic bacteria. PtNPs may be used to eliminate LPS that remains on the root surface, rendering the root surface innocuous to periodontal tissue. In addition, during endodontic treatment, it is difficult to kill bacteria that have entered the dentinal tubules. However, by using PtNPs in root canals, nanoparticles may enter the dentinal tubules and kill the bacteria. However, this study was only a limited basic scientific research effort, and further studies are necessary to determine the full potential of such nanoparticles for clinical applications. However, since this study is a limited basic study and a nonspecific reaction, further research is necessary for clinical application.

## Conclusion

PtNPs have been shown to mediate antibacterial effects related to caries, endodontic lesions, and periodontal diseases. PtNPs also exhibit functional potential to decompose proteins and strong effectiveness against LPS, the cell wall constituent of gram-negative bacteria. However, this study was limited to only *in vitro* findings, and there is a necessity to further investigate whether similar results can be obtained in the clinic.

## Supporting information

**S1 File. Fig 4 data.**
(XLSX)

**S2 File. Fig 6 data.**
(XLSX)

**S3 File. Fig 7 data.**
(XLS)

## Acknowledgments

The authors are grateful to Dr. Takuma Suzuki for assistance in measuring the amount of LPS.

## Author Contributions

**Conceptualization:** Hiroo Itohiya.

**Data curation:** Yuji Matsushima.

**Methodology:** Hiroo Itohiya, Yuji Matsushima, Sohtaro Kajiyama.

**Project administration:** Satoshi Shirakawa, Sohtaro Kajiyama.

**Software:** Satoshi Shirakawa.

**Supervision:** Akihiro Yashima, Takatoshi Nagano, Kazuhiro Gomi.

**Writing – original draft:** Hiroo Itohiya, Kazuhiro Gomi.

**Writing – review & editing:** Yuji Matsushima, Kazuhiro Gomi.

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
