## [Decision Letter · Decision Letter 0]

10 Jul 2019

PONE-D-19-16374

Organic resolution function and effects of non-colloidal platinum nanoparticles on bacteria and organic matter

PLOS ONE

Dear Professor Kazuhiro,

Thank you for submitting your manuscript to PLOS ONE. After careful consideration, we feel that it has merit but does not fully meet PLOS ONE’s publication criteria as it currently stands. Therefore, we invite you to submit a revised version of the manuscript that addresses the points raised during the review process.

We would appreciate receiving your revised manuscript by Aug 24 2019 11:59PM. To enhance the reproducibility of your results, we recommend that if applicable you deposit your laboratory protocols in protocols.io, where a protocol can be assigned its own identifier (DOI) such that it can be cited independently in the future. For instructions see: http://journals.plos.org/plosone/s/submission-guidelines#loc-laboratory-protocols

We look forward to receiving your revised manuscript.

Kind regards,

Amitava Mukherjee, ME, Ph.D.

Academic Editor

PLOS ONE

**Journal Requirements:**

2. At this time, we request that you please include in your methods section, or as supporting information, the full uncropped and un-altered blot and gel images that were used to make your figures. Please see the following link for more information about PLOS ONE guidlelines: http://journals.plos.org/plosone/s/submission-guidelines#loc-blots-and-gels.

3. Thank you for stating that “The funders had no role in study design, data collection and analysis, decision to publish, or preparation of the manuscript” in your financial disclosure.

Please also provide the name of the funders of this study (as well as grant numbers if available) in your financial disclosure statement.

**Comments to the Author**

1. Is the manuscript technically sound, and do the data support the conclusions?

Reviewer #1: Yes

Reviewer #2: No

2. Has the statistical analysis been performed appropriately and rigorously? 

Reviewer #1: Yes

Reviewer #2: Yes

3. Have the authors made all data underlying the findings in their manuscript fully available?

Reviewer #1: Yes

Reviewer #2: No

4. Is the manuscript presented in an intelligible fashion and written in standard English?

Reviewer #1: Yes

Reviewer #2: No

5. Review Comments to the Author

Reviewer #1: The submitted paper entitled" Organic resolution function and effects of non colloidal platinum nanoparticles on bacteria and organic matter" shows the antibacterial effects,its potential to decompose LPS and its use in peridontal applications. The manuscript is well written and deserves a publication with some minor revisions. Below is the some comments and suggestions to improve the quality of manuscript.

Technical comments

1. In material method section (Bacterial culture and regulation),write down the parameters for shaking at what rpm shaking was kept for aerobic bacteria

2. Add a section of stock solution preparation of NPs under material method

3. In material and method section "antibacterial testing" add a reference for the protocol utilized

4. In the discussion section, line no. 243,explain the mechanism of interaction of aerobic and anaerobic bacteria with NPs

5. In the discussion section, line no. 268, when LPS reacts with NPs is there any charge based interaction in Gram negative bacteria

6. Figure number 3 add an X axis demarcation

Reviewer #2: Authors reported non-colloidal platinum nanoparticles functional potential for decomposing organic matter. As the author pointed out, the NC-PtNPs increase fragmentation and decomposition of albumin, which may limit their practical applications because besides damging the LPs, the proposed NPs can be toxic to host through damaging the albumin. However, I found several serious problems in this manuscript and cannot accept this manuscript in a present form.

1. The authors claim non-colloidal form, but the particle size is 2–19 nm and suspended in water. Comment on it.

2. Although the NPs are obtained from commercial source, its size should be verified after suspended in water and data should be presented.

3. What will happen to fate of the NPs size and morphology after contacting either albumin or LPS.

4. What is mechanism of decomposition?

6. PLOS authors have the option to publish the peer review history of their article (what does this mean?). If published, this will include your full peer review and any attached files.

Reviewer #1: Yes: Dr. Ankita Mathur

Reviewer #2: No

---

## [Author Response · Author response to Decision Letter 0]

9 Aug 2019

Dear Reviewers,

We would like to thank you and the reviewers for the valuable comments on the original version of our manuscript. We have taken all these comments into account and submit, herewith, a revised version of our paper. We have addressed all the comments by reviewers, as indicated in the attached pages, and we hope that the explanations are satisfactory.

We hope that the revised version of our paper is now suitable for publication in the PLOS ONE, and we look forward to hearing from you at the earliest. 

Sincerely Yours,

Kazuhiro Gomi

Reviewer 1

1. Q: In material method section (Bacterial culture and regulation), write down the parameters for shaking at what rpm shaking was kept for aerobic bacteria.

→ A: I changed the description as follows according to your indication.

P6, line 96, 99 as follow.

Streptococcus mutans was cultured in Tryptic Soy Broth (TSB, FUJIFILM Wako Pure Chemical Co., Osaka, Japan) and agar medium (Becton Dickinson) with 4% glucose (FUJIFILM Wako Pure Chemical Co.) at 37 ℃ under aerobic conditions for 2 days, followed by preculture in TSB liquid medium with 4% glucose under aerobic shaking at 100 rpm in shaking incubator (SI-782, Optima Inv. Tokyo, Japan) for 2 days. Enterococcus faecalis was cultured in TSB agar medium (at 37 ℃ under aerobic conditions for 2 days), followed by preculture in TSB liquid medium under aerobic shaking at 100 rpm in shaking incubator for 2 days.

2. Q: Add a section of stock solution preparation of NPs under material method.

→ A: We added the information about preparation of PtNPs and stock solution in P5, line 73-780 as follow.

In this experiment, 100 ppm PtNPs (particle size: 2–19 nm) solution, which was used as a stock solution, were obtained from the manufacturer (Ceramics Craft Co. Ltd. Shizuoka, Japan). The PtNPs were made directly from platinum by irradiation with infrared pulsed laser in liquid according to the Liquid phase laser ablation method. The storage of PtNPs was kept in the dark at room temperature avoiding direct sunlight as the stock solution. Moreover, when using for experiment, the stock solution was diluted in sterile purified water and used each time. Fig.1 shows a transmission electron micrograph of PtNPs used in this experiment.

3. Q: In material and method section "antibacterial testing" add a reference for the protocol utilized.

→ A: We added the reference, which is show in below, in P7 line 111

Ref. 25

Decreased Phototoxic Effects of TiO₂ Nanoparticles in Consortium of Bacterial Isolates from Domestic Waste Water

Ankita Mathur, Jyoti Kumari, Abhinav Parashar, Lavanya T., N. Chandrasekaran, Amitava Mukherjee 

Published: October 23, 2015https://doi.org/10.1371/journal.pone.0141301

4. Q: In the discussion section, line no. 243, explain the mechanism of interaction of aerobic and anaerobic bacteria with NPs.

→ A: We added the explanation of the mechanism of interaction of aerobic and anaerobic bacteria with NPs in P15 line 255-P16 line 259.

While, it is inferred that the antibacterial effect of NPs on anaerobic bacteria of oral origin is lower than that on aerobic bacteria. It is considered that this is because the antibacterial property is controlled by the availability of oxygen and the particle size34). These interactions may have shown similar antibacterial activity to P. gingivalis, S. mutans and E. faecalis.

34): Size-dependent antibacterial activities of silver nanoparticles against oral anaerobic pathogenic bacteria. Lu Z, Rong K, Li J, Yang H, Chen R. J Mater Sci Mater Med. 2013 Jun;24(6):1465-71. doi: 10.1007/s10856-013-4894-5. Epub 2013 Feb 26.

5. Q: In the discussion section, line no. 268, when LPS reacts with NPs is there any charge based interaction in Gram negative bacteria.

→ A: We added the following in P17 line 283-286.

LPS, which is a component of the cell wall of gram negative bacteria, is negatively charged and has a high affinity to NPs and binds tightly. And it is thought that destruction of LPS arises from LPS and NPs from a physical interaction37）.

37）Metal nanoparticles: understanding the mechanisms behind antibacterial activity. Slavin YN, Asnis J, Häfeli UO, Bach H. J Nanobiotechnology. 2017 Oct 3;15(1):65. doi: 10.1186/s12951-017-0308-z.

6. Q: Figure number 3 add an X axis demarcation.

→ A: We add to X axis demarcation in Fig 3 (change to 4).　 

Reviewer 2

0. Q: As the author pointed out, the NC-PtNPs increase fragmentation and decomposition of albumin, which may limit their practical applications because besides damging the LPs, the proposed NPs can be toxic to host through damaging the albumin.

P19, line 306-307

→ A: Following your suggestions, I added the following sentence.

However, since this study is a limited basic study and a non-specific reaction, further research is necessary for clinical application

1. Q: The authors claim non-colloidal form, but the particle size is 2–19 nm and suspended in water. Comment on it.

→ A: This PtNPs were made directly from platinum by irradiation with infrared pulsed laser. However, we don't have detailed data about non-colloidal or colloidal. Therefore, I would like to revise this paper with the term platinum nanoparticles (PtNPs), rather than using the term non-colloidal platinum nanoparticles (NC-PtNPs).

Thank you for suggesting very important points.

2. Q: Although the NPs are obtained from commercial source, its size should be verified after suspended in water and data should be presented.

P5, line 82-84

→ A: Following your suggestions, we have added an electron micrograph of PtNPs.

3. Q: What will happen to fate of the NPs size and morphology after contacting either albumin or LPS.

P14, line 205-210 

→ A: Following your suggestions, we have added an electron micrograph of PtNPs with albumin or LPS.

4. Q: What is mechanism of decomposition?

→ A: Added the following comments regarding the mechanism of disassembly.

 P18, line 291-293

Albumin and LPS in contact with PtNPs are thought to be reduced in molecular weight due to cleavage of the protein backbone due to the oxidative degradation reaction or physical interaction caused by the generated ROS38）. However, the detailed mechanism is still unknown.

38）Mol Aspects Med. 2014 Feb;35:1-71. doi: 10.1016/j.mam.2012.09.001. Epub 2012 Oct 26.

Protein damage, repair and proteolysis.

Chondrogianni N, Petropoulos I, Grimm S, Georgila K, Catalgol B, Friguet B, Grune T, Gonos ES.

---

## [Decision Letter · Decision Letter 1]

5 Sep 2019

[EXSCINDED]

Organic resolution function and effects of platinum nanoparticles on bacteria and organic matter

PONE-D-19-16374R1

Dear Dr. Kazuhiro,

We are pleased to inform you that your manuscript has been judged scientifically suitable for publication and will be formally accepted for publication once it complies with all outstanding technical requirements.

With kind regards,

Amitava Mukherjee, ME, Ph.D.

Academic Editor

PLOS ONE

Additional Editor Comments (optional):

Reviewers' comments:

Reviewer's Responses to Questions

**Comments to the Author**

1. If the authors have adequately addressed your comments raised in a previous round of review and you feel that this manuscript is now acceptable for publication, you may indicate that here to bypass the “Comments to the Author” section, enter your conflict of interest statement in the “Confidential to Editor” section, and submit your "Accept" recommendation.

Reviewer #1: All comments have been addressed

Reviewer #2: All comments have been addressed

2. Is the manuscript technically sound, and do the data support the conclusions?

Reviewer #1: Yes

Reviewer #2: Yes

3. Has the statistical analysis been performed appropriately and rigorously? 

Reviewer #1: Yes

Reviewer #2: Yes

4. Have the authors made all data underlying the findings in their manuscript fully available?

Reviewer #1: Yes

Reviewer #2: Yes

5. Is the manuscript presented in an intelligible fashion and written in standard English?

Reviewer #1: Yes

Reviewer #2: Yes

6. Review Comments to the Author

Reviewer #1: All the questions that were raised while reviewing had been addressed and thereby paper could be published.

Reviewer #2: The authors considerably improved the manuscript titled: Organic resolution function and effects of platinum nanoparticles on bacteria and organic matter. It may be acceptable

7. PLOS authors have the option to publish the peer review history of their article (what does this mean?). If published, this will include your full peer review and any attached files.

Reviewer #1: No

Reviewer #2: Yes: Anbazhagan Veerappan

---

## [Editor Report · Acceptance letter]

12 Sep 2019

PONE-D-19-16374R1 

Organic resolution function and effects of platinum nanoparticles on bacteria and organic matter 

Dear Dr. Gomi:

I am pleased to inform you that your manuscript has been deemed suitable for publication in PLOS ONE. Congratulations! Your manuscript is now with our production department. 

With kind regards,

on behalf of

Professor Dr. Amitava Mukherjee 

Academic Editor

PLOS ONE